https://doi.org/10.5194/egusphere-2024-2500



# Enhanced understanding of atmospheric blocking modulation on ozone dynamics within a high-resolution Earth system model

Wenbin Kou[1], Yang Gao[1]*, Dan Tong[2], Xiaojie Guo[3,4], Xiadong An[5], Wenyu Liu[2], Mengshi Cui[2], Xiuwen Guo[1], Shaoqing Zhang[6], Huiwang Gao[1], Lixin Wu[6]

[1]Frontiers Science Center for Deep Ocean Multispheres and Earth System and Key Laboratory of Marine Environmental Science and Ecology, Ministry of Education, Ocean University of China, and Laoshan Laboratory, Qingdao, 266100, China

[2]Department of Earth System Science, Tsinghua University, Beijing, 100084, China

[3]International Center for Climate and Environment Sciences, Institute of Atmospheric Physics, Chinese Academy of Sciences, Beijing, 100029, China

[4]University of Chinese Academy of Sciences, Beijing, 100049, China

[5]College of Oceanic and Atmospheric Sciences, Ocean University of China, Qingdao, 266100, China

[6]Frontiers Science Center for Deep Ocean Multispheres and Earth System, and Key Laboratory of Physical Oceanography, Ministry of Education, the College of Oceanic and Atmospheric Sciences, Ocean University of China, and Laoshan Laboratory, Qingdao, 266100, China

*Correspondence to: yanggao@ouc.edu.cn





## Abstract

High concentrations of surface ozone pose significant health risks, yet understanding the factors governing ozone levels, particularly the influence of large-scale circulations, remains incomplete. A key challenge lies in accurately modeling both large-scale circulations and ozone concentrations. Leveraging recent advancements in optimizing a high-resolution Earth system model with 25 km atmospheric resolution, how local meteorology and large-scale circulations impact ozone concentrations is investigated. We find that heatwaves can trigger substantial increases in ozone concentrations by stimulating biogenic volatile organic compound (BVOC) emissions during the summers of 2015-2019. For example, compared to non-heatwave periods, ozone concentrations during heatwaves increase by 12.0 ppbv in the southeastern U.S., 9.7 ppbv in Europe, 17.6 ppbv in North China, and 9.0 ppbv in central eastern China. In addition to local effects, atmospheric blocking strongly influences downstream meteorological conditions and ozone formation. Focusing on ozone pollution in eastern China, we identify three major pathways of Rossby wave propagation based on blocking locations: the Euro-Atlantic sector, northern Russia, and the North Pacific, inducing increased air temperature and intensified downward surface solar radiation downstream. The impact of blocking is most pronounced over central eastern China, where ozone concentrations during blocking increase by 5.9 ppbv to 10.7 ppbv compared to reference periods, followed by North China, ranging from 2.1 ppbv to 4.9 ppbv. Blocking can stimulate more BVOC emissions, enhancing ozone concentrations by 10.6 ppbv to 15.9 ppbv. These findings underscore the critical role that large-scale atmospheric circulation patterns play in regional-scale air quality, particularly under a warming climate.

Key words: atmospheric blocking, ozone, Rossby wave propagation, BVOC emissions



## Summary


Unlike traditional numerical studies, we apply a high-resolution Earth system model,
improving simulations of ozone and large-scale circulations such as atmospheric
blocking. In addition to local heatwave effects, we quantify the impact of atmospheric
blocking on downstream ozone concentrations, which is closely associated with the
blocking position. We identify three major pathways of Rossby wave propagation,
stressing the critical role of large-scale circulation play in regional air quality.



## 1. Introduction

Air pollution ranks as the fourth leading global risk factor for mortality, trailing high systolic blood pressure, tobacco use, and dietary risks (Brauer et al., 2021). Among atmospheric pollutants, ambient ozone is a major contributor to this burden (Fuller et al., 2022), affecting human health (Nuvolone et al., 2018), global climate (Deitrick and Goldblatt, 2023), and ecosystem health through exacerbating crop yield losses (Emberson et al., 2018).

The HTAP (Hemispheric Transport of Air Pollution; Dentener et al., 2010; Parrish et al., 2012) and TOAR (Tropospheric Ozone Assessment Report; Tarasick et al., 2019) programs have extensively studied long-term ozone trends. Their synthesis in 2021 (Parrish et al., 2021b) reveals a twofold increase in lower tropospheric ozone at northern mid-latitudes from 1950 to 2000. The World Health Organization (WHO) strengthened air quality standards in 2021, emphasizing the critical need to assess ozone trends and their key drivers.

Ozone, a secondary air pollutant, forms when emission precursors such as volatile Organic Compounds (VOCs) and NOx are present (Fu and Tian, 2019). While anthropogenic emissions are significant, biogenic VOC (BVOC) emissions, which comprise about 90% of global VOC emissions (Guenther et al., 2012), are particularly sensitive to temperature. For instance, BVOC emissions notably elevate surface ozone levels in the North China Plain, contributing to increases of 7.8 ppbv and 10.0 ppbv in the regional average MDA8 ozone concentrations in the North China Plain and Beijing, respectively, during the summer of 2017 (Ma et al., 2019). Even in less polluted regions such as the U.S., BVOC emissions contribute a notable fraction of ozone, averaging 10% and 19% in the western and southeastern U.S., respectively (Zhang et al., 2017).

This effect is amplified under favorable meteorological conditions. Compared to non-heatwave periods, heatwaves trigger increased BVOC emissions, resulting in regional daytime ozone concentration increases of 10 µg m$^{-3}$ in the Pearl River Delta, with peaks reaching 42.1 µg m$^{-3}$ (Wang et al., 2021). In southwestern Europe, heatwaves induce a 33% rise in BVOC emissions, resulting in surface ozone



concentration increases of 9 µg m$^{-3}$ during the summers of 2012-2014 (Guion et al.,
2023). However, biases in modeling heatwaves (Gao et al., 2012) and ozone, such as
overestimations up to 20 ppbv in low-resolution global models (Emmons et al., 2020;
Lamarque et al., 2012), have hindered previous investigations, primarily conducted
using regional weather and chemistry models (Gao et al., 2020; Zhang et al., 2022).
Addressing these challenges, especially the biases from low-resolution global models
in boundary conditions (Zeng et al., 2022), is crucial for advancing Earth system models
to better understand the impact of heatwaves on ozone through BVOC emissions.
Local meteorological factors, particularly high temperatures, are closely linked to
large-scale circulations (Li and Sun, 2018), which further influence the ozone-
temperature relationship. For instance, the correlation between summer surface ozone
and temperature over eastern North America correlates with the position of the jet
stream, defined by the latitude of the maximum 500 hPa zonal wind averaged across
the region (Barnes and Fiore, 2013). Atmospheric circulations, such as the North
Atlantic Oscillation, significantly affect moisture transport, precipitation, and
subsequently, trace gas transport, deposition and air pollutant concentrations
(Christoudias et al., 2012). In  central eastern China, the East Asian summer monsoon
explains 2%-5% of interannual variations in surface ozone concentrations (Yang et al.,
2014). Moreover, a positive phase of the Eurasian teleconnection induces Rossby wave
train propagation from Europe to North China, influencing downward surface solar
radiation intensity and temperatures, thereby modulating ozone concentration
variability (Yin et al., 2019).
Recently, Yang et al. (2022) highlighted that high temperatures alone may not
always enhance ozone formation.. For instance, high temperatures induced by a zonal
'+ - +' wave-train pattern over Eurasia at 300 hPa may not favor ozone enhancement in
North China.   In contrast, circulation anomalies resembling an atmospheric blocking
pattern, including positive geopotential height anomalies at 300 hPa over North China
and eastern Eurasia, can lead to weaker meridional temperature gradients, intensified
downward solar radiation, reduced cloud cover, and aggravated ozone pollution.



Atmospheric blocking, a quasi-stationary, large-scale extra-tropical weather system,
often occurs over expansive regions like the North Atlantic-Europe and North Pacific
(Pelly and Hoskins, 2003; Schwierz et al., 2004; Woollings et al., 2018). Blocking highs
are frequently associated with extreme weather events (Barriopedro et al., 2011;
Cattiaux et al., 2010). For example, through downstream Rossby wave propagation
from Alaska to East Asia, Alaska blocking can induce subsequent blocking over the
Urals, influencing extreme cold events across North America and Eurasia (Yao et al.,

2023).

Despite significant advancements, the impact of atmospheric blocking on extreme
weather events and ozone remains insufficiently explored. For example, using a
Hovmöller diagram and local wave activity calculated from 500 hPa geopotential height,
Sun et al. (2019) found that variations in wave activity can explain 30-40% of ozone
variability in historical U.S. summers. Challenges in global models, such as simulated
biases in atmospheric blocking and ozone, including overestimations (Clifton et al.,
2020), have undermined confidence in linking large-scale circulation patterns with
ozone levels (Barnes and Fiore, 2013).
Building on recent advances in high-resolution Earth system models that mitigate
ozone biases (Gao et al., in review-b) and simulate meteorological parameters and
climate extremes (Chang et al., 2020; Gao et al., in review-a; Gao et al., 2023; Guo et
al., 2022), this study is structured as follows. Section 2 describes the model setup. It is
followed by an analysis of observational ozone data, the effects of BVOC emissions,
and heatwaves on ozone concentrations. Finally, we explore how atmospheric blocking
influences ozone pollution in eastern China.

**2 Method and data**
**2.1 Model configurations**
In this study, we utilize the Community Earth System Model version 1.3,
employing the Community Atmosphere Model 5.0 (CAM5) as its atmospheric
component. CAM5 runs at two spatial resolutions: nominal 1° and 0.25°. Sea surface





temperature and sea ice are prescribed at a spatial resolution of of $1.0° \times 1.0°$.
Atmospheric gas chemistry and aerosol processes are simulated using the Model for
OZone And Related chemical Tracers (MOZART) and the three-mode version of the
Modal Aerosol Module (MAM3). The high-resolution and low-resolution
configurations of CESM are denoted as SW-HRESM and CESM-LR, respectively.
Further details can be found in Gao et al. (in review-b). The simulation period covers
June to August from 2015 to 2019, with May used for spin-up to mitigate initial
condition influences.

Emissions for the simulations are sourced as follows: anthropogenic emissions

from the Copernicus Atmosphere Monitoring Service global emissions (CAMS-
GLOB-ANT v4.2-R1.1; Granier et al., 2019), with updates for China based on the
Multi-resolution Emission Inventory for China (MEIC; Li et al., 2017). Volcanic
emissions are from Global Emission Inventory Activity (GEIA), and aircraft emissions
from the Community Emission Data System (CEDS). Biomass burning emissions data
are sourced from the Fire INventory from National Center for Atmospheric Research
(FINN) version 2.5 (Wiedinmyer et al., 2023). High-resolution simulations use
emissions data at 0.1° resolution, while low-resolution simulations aggregate emissions
from 0.1° to ~1.0° resolution. Biogenic emissions are calculated online using the Model
of Emissions of Gases and Aerosols from Nature version 2.1 (MEGAN2.1; Guenther et
al., 2012). Further emission details are available in Gao et al. (in review-b).

Two numerical experiments are designed to assess the impact of BVOC emissions

on ozone. The first experiment includes all emissions (BASE case), while the second
experiment turns off BVOC emissions (No_BVOC case). By subtracting results from
the No_BVOC case from those of the BASE case, we isolate the contribution of BVOC
emissions to ozone.

**2.2 Blocking detection method and Rossby wave flux calculation**

To identify atmospheric blocking, we use a two-dimensional hybrid blocking index

based on 500 hPa geopotential height. The index is applied across a range of latitudes,



$\phi$ , (40° to 75° N) for each longitude, $\lambda$ , incorporating meridional gradients to
identify blocked grid points:

$$GHGN(\lambda,\phi) = \frac{Z(\lambda,\phi+\Delta) - Z(\lambda,\phi)}{\Delta} < -10$$

,

$$GHGS(\lambda,\phi) = \frac{Z(\lambda,\phi) - Z(\lambda,\phi-\Delta)}{\Delta} > 0$$

,

$$Z_{\text{anomaly}}(\lambda,\phi) = Z(\lambda,\phi) - \bar{Z}(\phi) > 0,$$


where, GHGN (GHGS) indicates the meridional gradient to the north (south) of
geopotential height at 500 hPa, Z means the 500 hPa geopotential height at longitude
$\lambda$ along latitude $\phi$ , and $\bar{Z}$ is the zonal (0° to 360°) average of Z at latitude $\phi$ ; $\Delta$ is
set as 15°.

A blocking region is defined when the meridional extension of blocked grid points
exceeds 15°. The center of each blocking region is determined as the grid point with
maximal 500 hPa geopotential height. Sequential blocking events are identified if the
center of a blocking region on one day was within a specified distance (27° in latitude
× 36° in longitude) of the center on the previous day. We restrict a blocking event lasting
at least five days. More information can be found in Masato et al. (2013) and Gao et al.
(in review-a).
To examine Rossby wave propagation, the horizontal stationary wave activity flux
(W) is calculated following Takaya and Nakamura (2001). Key variables used for flux
calculation include zonal wind (U), meridional wind (V), wind speed ($|\mathbf{U}|$), and
anomalous geopotential height ($\Phi'$).

$$\mathbf{W} = \frac{P\cos\phi}{2|\mathbf{U}|} \cdot \begin{pmatrix} \frac{U}{a^2\cos^2\phi}\left[\left(\frac{\partial\psi'}{\partial\lambda}\right)^2 - \psi'\frac{\partial^2\psi'}{\partial\lambda^2}\right] + \frac{V}{a^2\cos\phi}\left[\frac{\partial\psi'}{\partial\lambda}\frac{\partial\psi'}{\partial\phi} - \psi'\frac{\partial^2\psi'}{\partial\lambda\partial\phi}\right] \\ \frac{U}{a^2\cos\phi}\left[\frac{\partial\psi'}{\partial\lambda}\frac{\partial\psi'}{\partial\phi} - \psi'\frac{\partial^2\psi'}{\partial\lambda\partial\phi}\right] + \frac{V}{a^2}\left[\left(\frac{\partial\psi'}{\partial\phi}\right)^2 - \psi'\frac{\partial^2\psi'}{\partial\phi^2}\right] \end{pmatrix}, \quad (1)$$

where $\mathbf{W}$ represents the wave activity flux (unit: m² s⁻²), $\psi'$ ($= \Phi'/f$) represents the
geostrophic stream function, f ($= 2\Omega\sin\phi$) is the Coriolis parameter, P is the normalized
pressure (P per 1000 hPa), and a is Earth's radius. λ and ϕ denote the longitude and



latitude, respectively.

**2.3 Observational data**

Observational ozone data are collected from several platforms, including the Air

Quality System (AQS, https://www.epa.gov/aqs; last access: 30 June, 2023) and the
Clean Air Status and Trends Network (CASTNET, https://www.epa.gov/castnet; last
access: 30 April, 2023) in the U.S., the European Monitoring and Evaluation
Programme database (EMEP; http://ebas.nilu.no; last access: 30 January 2023) in
Europe,  and the China National Environmental Monitoring Center (CNEMC,
http://www.pm25.in; last access: December 8, 2021) in China. The monitoring network
comprises 1293 sites for AQS, 99 for CASTNET, 286 for EMEP and 2025 for CNEMC.
Meteorological data used in this study are sourced from the National Centers for
Environmental Prediction's Reanalysis-1 (NCEP; Kalnay et al., 1996).

**3 Results and discussion**
**3.1 Characteristics of observed ozone in the Northern Hemisphere**

Fig. 1 illustrates the characteristics of observed ozone levels based on a

comprehensive analysis of extensive observational datasets. Peak season ozone (Fig.
1a), as defined by the WHO in 2021, is determined using a 6-month running average of
maximum daily 8-h (MDA8) ozone concentrations for each grid, with the maximum
value being considered. The WHO air quality guideline is set at 60 $\mu g \cdot m^{-3}$ (31 ppbv),
with additional standards of 100 $\mu g \cdot m^{-3}$ (51 ppbv) and 70 $\mu g \cdot m^{-3}$ (36 ppbv). Regional
differences in ozone pollution are apparent: higher concentrations are observed in the
western U.S. due to elevated altitude and background levels (Parrish et al., 2021a).
Specific sites with significant ozone pollution include L.A. and Houston, as previously
documented (Dunker et al., 2017). In Europe, ozone pollution is more pronounced in
southern regions, particularly around the Mediterranean, consistent with earlier studies
(Zohdirad et al., 2022). In China, the eastern region exhibits concentrated pollution.
Mean peak season ozone levels are 45.5 ppbv in the U.S., 42.9 ppbv in Europe, and



53.7 ppbv in China.

The cumulative distribution function of peak season ozone concentrations is

shown in Fig. 1b using gridded data. In the U.S. and Europe, only 15% and 8% of the
peak season ozone concentrations, respectively, exceed the level I (51ppbv) from 2015
to 2019, whereas in China, almost 60% exceed this threshold. However, when applying
the stricter standard (36ppbv), exceedance rates are notably high: 98%, 89%, and 96%
in the U.S., Europe and China, respectively.

Fig. 1c presents the fourth highest MDA8 ozone values annually from 2015 to

2019, alongside daily values for the U.S., Europe and China. The WHO has established
standards at 82 ppbv and 61 ppbv, with an air quality guideline of 51ppbv. Exceedance
rates for the strictest guideline (51 ppbv) are 98%, 89% and 99% in the U.S., Europe
and China, respectively. For the 61ppbv standard, rates are 78%, 60% and 96%,
respectively, and for the 82ppbv standard, exceedance rates are 2%, 4% and 77%,
respectively. Considering all daily values, with a sample size approximately 365 times
larger than the annual fourth highest value, the rates of ozone exceedance (i.e.,
exceeding 51 ppbv) are observed to be 17% in the U.S., 11% in Europe, and 31% in
China. This indicates that there are significantly more days where ozone levels exceed
the threshold beyond just the fourth highest maximum daily 8-hour (MDA8) ozone
level in these regions. This suggests that air quality issues related to ozone are more
persistent and widespread than what might be inferred solely from the fourth highest
MDA8 metric.



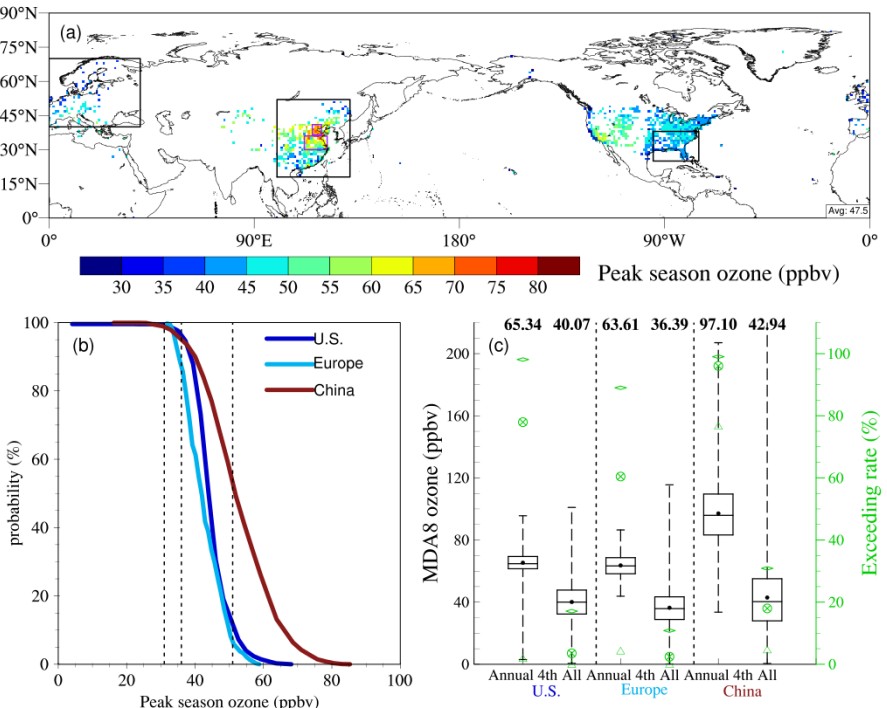

**Fig. 1 Peak season ozone concentrations and maximum daily 8-hr (MDA8) ozone concentrations.** (a) Spatial distribution of mean peak season ozone concentrations in the Northern Hemisphere from 2015 to 2019. The black squares represent regions in Europe, eastern China, and the U.S., while the purple squares in eastern China denote North China and central eastern China regions. (b) Cumulative Distribution Function of peak season ozone concentrations, with dashed lines indicating WHO standard values (31 ppbv, 36 ppbv, 51 ppbv) set by WHO. (c) Box-and-whisker plot of annual fourth-highest (left) and all (right) MDA8 during 2015-2019 in the U.S., Europe and China. The boxes represent the interquartile range (25th to 75th percentiles), horizontal lines denote medians, solid points indicate averages, and line end points show maximum and minimum values. Exceedance rates (%) of MDA8 to WHO standards of 82 ppbv, 61 ppbv, and 51 ppbv are marked with green triangle, crossed-out circle, and diamond symbols, respectively.



**3.2 BVOC emissions and their effects on ozone**

BVOC emissions during the summer months of 2015-2019 are depicted in Fig. 2a and Fig. 2b, with global totals of 86.0 Tg month$^{-1}$ in SW-HRESM and 90.7 Tg month$^{-1}$ in CESM-LR. Isoprene emissions (Fig. 2c, d) account for nearly half of these totals amounting to 42.3 Tg month$^{-1}$ in SW-HRESM and 45.2 Tg month$^{-1}$ in CESM-LR. This predominance of isoprene emissions aligns with previous studies (Ma et al., 2022; Mochizuki et al., 2020). Isoprene emissions are predominantly concentrated in tropical regions, reflecting the prevalence of dense forest cover. Our study indicates values approximately 30% higher than those (Fig. S1) reported in Weng et al. (2020) due to previously underestimated emissions in tropical regions.

To assess the utility of high-resolution simulations, we compute the standard deviation across 16 grid points in SW-HRESM corresponding to a single low-resolution grid (Fig. S2). The average monthly isoprene emissions during 2015-2019 are 0.63 kg m$^{-2}$, 0.51 kg m$^{-2}$ and 0.21 kg m$^{-2}$ over the U.S., Europe and China (Fig. 2c), respectively, with mean standard deviation of 0.13 kg m$^{-2}$, 0.11 kg m$^{-2}$, 0.05 kg m$^{-2}$ (Fig. S2). This ratio also applies to biogenic emission-rich areas such as the southeastern U.S., southern Europe and eastern China, highlighting the importance of using finer grid spacings for accurately capturing the spatial heterogeneity of BVOC emissions.

The spatial distribution of BVOC emissions closely correlates with the distribution of broadleaf trees (Fig. S3), which have higher emission factors compared to other plant types (Table 2 in Guenther et al., 2012). Isoprene emissions are most intense in tropical regions where broadleaf evergreen and deciduous tropical trees predominate, as well as in mid-to-high latitude belts and isolated hotspots in mid-latitudes like the southeastern U.S., southern Europe, and eastern China.

An exception is observed in the Amazon region, where despite dense broad evergreen tropical forest cover, the largest isoprene and BVOC emissions occur away from the main forest area. This Amazon hotspot, noted in previous studies (Opacka et al., 2021), is influenced by key meteorological factors such as 2-meter air temperature and downward surface solar radiation (Fig. S4). Specifically, areas with higher



temperatures and stronger solar radiation exhibit greater BVOC and isoprene emissions.
The discrepancy in temperature between CESM-LR and SW-HRESM simulations
reveals nuances in emission patterns, with CESM-LR showing slightly higher
temperatures that lead to increased emissions. The slightly lower temperature in higher
grid spacing simulations in regional climate model was also reported by Pugh et al.
(2013). They suggested that improved representation of forests could increase latent
heat flux and thereby mitigate temperature rises through a reduced sensible heat. The
study compared three grid spacings: 0.1°, 0.5°, and 2.0°, showing that across regions
such as South America, Southeast Asia, and the southeastern U.S., there was a small
overall difference of about 2% in BVOC emissions on a regional scale. However, this
difference could reach up to 150% in high-emission areas.

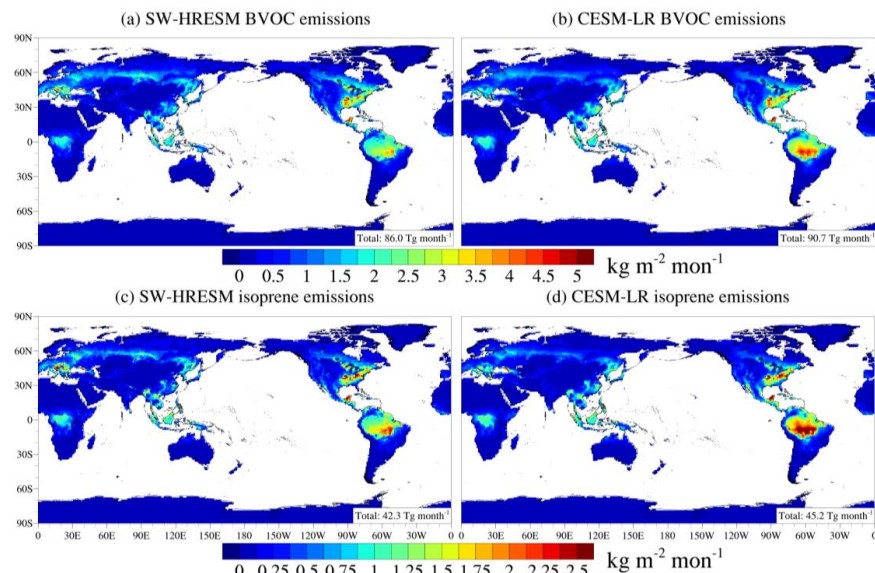


**Fig. 2 Spatial distribution of BVOC (top) and isoprene (bottom) emissions based**
**on SW-HRESM (left) and CESM-LR (right).** Shown are monthly total emissions
averaged during the summer of 2015-2019.

To understand the contribution of BVOC emissions to ozone concentrations across
different grid resolutions, we compare two scenarios: one with biogenic emissions





included and one without. Fig. 3a and 3b illustrate the spatial distribution of ozone
concentrations averaged over the summers of 2015-2019 for both SW-HRESM and
CESM-LR. Both models identify significant ozone pollution areas in the Northern
Hemisphere, particularly over southern Europe, the southeastern U.S., and eastern
China. The contribution of BVOC emissions to ozone concentrations is further detailed
in Fig. 3c-f.

In SW-HRESM, BVOC emissions contribute approximately 2.2 ppbv to the global

mean ozone concentrations over land, representing 7% relative to the mean value of
31.3 ppbv (Fig. 3c,e). However, the impact of BVOC emissions on ozone
concentrations is modulated by factors such as anthropogenic emissions and
meteorological conditions. Regions with abundant BVOC emissions and higher ozone
concentrations, such as the U.S., Europe, and eastern China, show a substantial
contribution of 15% to 30% from BVOC emissions to ozone levels. In contrast, the
Amazon rainforest in Brazil, despite having the highest BVOC emissions, exhibits a
negative contribution to ozone levels. This is attributed to the fact that in regions with
low NOx concentrations, increased VOCs initiated by OH oxidation can lead to the
formation of stable organic nitrogen compounds, through increasing organic peroxy
radicals and elevating the reaction with $NO_2$ (Tonnesen and Jeffries, 1994). It reduces
the availability of $NO_2$ and the subsequent photolysis such as a reduction of O3P,
thereby reducing ozone concentrations (Kang et al., 2003; Unger, 2014). While this
effect is evident in CESM-LR, lower resolution simulations may overlook finer-scale
variability, affecting the accuracy of quantifying the impact of BVOC emissions on
ozone.






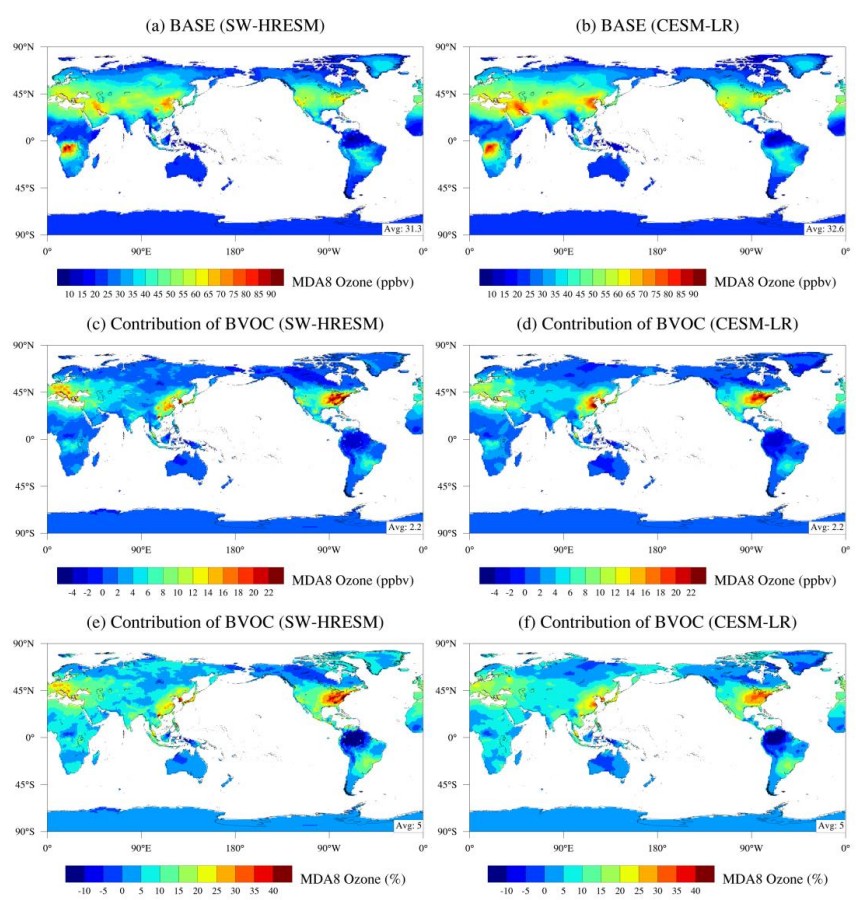

**Fig. 3 Spatial distribution of MDA8 ozone from SW-HRESM (left) and CESM-LR (right).** Shown are results of ozone concentrations at BASE (top) and the contribution of BVOC emissions to ozone (middle row: ppbv; bottom row: %). Global mean values over land are indicated in the bottom right.

### 3.3 Effects of heatwaves on ozone

Heatwaves not only accelerate photochemical reactions but also intensify BVOC emissions, thereby amplifying ozone production and exacerbating ozone pollution. Building on previous studies (Gao et al., 2012; Sillmann et al., 2013), heatwaves are defined within each grid as periods when the daily mean near-surface air temperature exceeds the 90th percentile of the climatological mean, focusing on the summer period





from 2015 to 2019 in this study. To quantify the impact of heatwaves on ozone
concentrations, Fig. 4 illustrates the probability distribution function (PDF) of MDA8
ozone concentrations for both the BASE case and a scenario without BVOC emissions,
aggregated across entire summer periods and specifically during heatwave days. Given
the superior capability of high-resolution simulations in reproducing heatwaves and
ozone concentrations (Gao et al., in review-b; Gao et al., 2023), we present results
solely from SW-HRESM hereafter.

Several notable observations emerge. Firstly, a comparison of heatwave periods to

non-heatwave periods (solid red vs. solid blue lines in Fig. 4) reveals a noticeable
rightward shift in the PDF, indicating an increase in ozone levels due to heatwave
impacts, a well-established phenomenon (e.g., Gao et al., 2020; Zhang et al., 2018).
Specifically, compared to non-heatwave periods, mean ozone concentrations increase
by 9.1 ppbv, 9.7 ppbv, and 8.4 ppbv during heatwaves over the U.S., Europe, and eastern
China, respectively. This effect is more pronounced in specific regions, such as North
China (NC) with an increase of 17.6 ppbv, followed by the southeastern U.S. (12.0 ppbv)
and central eastern China (CECN) (9.0 ppbv), accounting for 12% to 21% of regional
mean ozone levels. A previous study noted that median surface ozone concentrations
during U.S. heatwaves from 1990 to 2016 could increase by 10% to 80% (Meehl et al.,

2018).

Comparing scenarios with and without BVOC emissions (solid vs. dashed lines in

Fig. 4), BVOC emissions significantly contribute to ozone enhancement during both
non-heatwave and heatwave periods. For instance, during heatwaves, BVOC emissions
contribute 20.9 ppbv, 10.4 ppbv, 14.4 ppbv, and 20.5 ppbv over the southeastern U.S.,
Europe, North China, and central eastern China, respectively. A study by Churkina et
al. (2017) found that biogenic emissions contributed 17-20% to ozone formation in
Berlin, Germany, in July 2006, with this contribution potentially increasing to 60%
during heatwaves.

It is important to note that the influence of BVOC emissions persists outside of

heatwave periods, particularly when downward surface solar radiation remains





sufficiently high (Fig. S5). The differences in BVOC contributions to ozone between
heatwave and non-heatwave periods represent the incremental effect of BVOCs during
heatwaves, accounting for 7.7 ppbv, 3.9 ppbv, 2.0 ppbv, and 6.7 ppbv over these four
regions, respectively. This incremental effect constitutes 64%, 40%, 11%, and 74% of
the total heatwave effects, indicating varying degrees of BVOC influence across
different regions. The relatively smaller incremental BVOC effect during heatwaves
over North China is partly attributed to higher anthropogenic emissions and lower
BVOC emissions compared to the other regions. With potential reductions in
anthropogenic emissions in China, BVOC emissions could assume a more pivotal role,
especially given projections of increased frequency of heatwaves in a warming climate
(Gao et al., 2023; Gao et al., 2022).

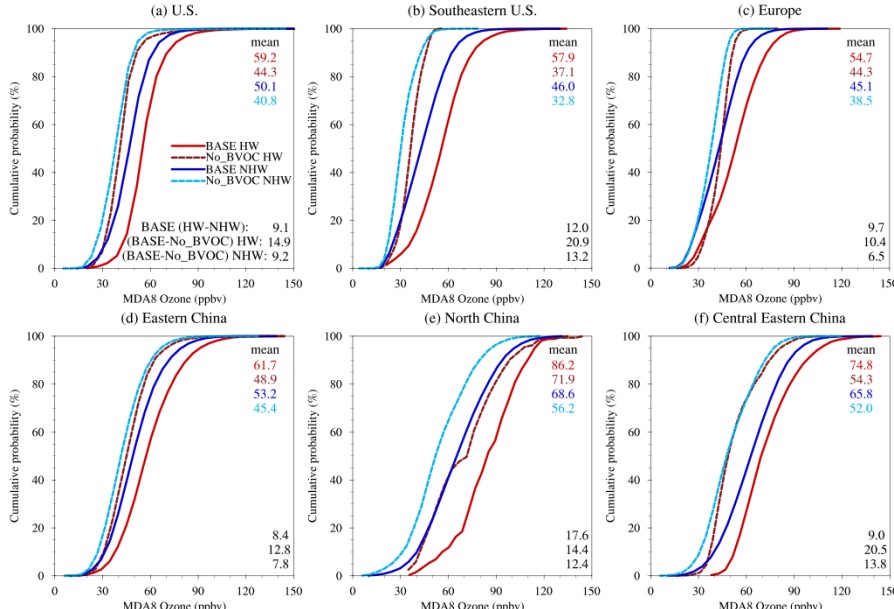


**Fig. 4 Cumulative Density Function (CDF) of MDA8 ozone concentrations.** Shown
are results for the BASE case (solid line) and the case without BVOC emissions (dashed
line), during heatwaves (red) and non-heatwaves (blue) based on SW-HRESM.










**3.4 The role of atmospheric blocking on ozone pollution in eastern China**

Eastern China has emerged as a significant region grappling with severe ozone pollution. Numerous studies have endeavored to explore the driving factors, particularly in the last decade, leveraging the widespread availability of ozone data across China. For example, through the examination of ozone pollution events in North China during 2014-2017, Gong and Liao (2019) investigated ozone pollution episodes in North China from 2014 to 2017 and identified that under weather conditions characterized by high near-surface air temperatures, low relative humidity, and anomalous southerly winds in the lower troposphere, ozone concentrations tend to accumulate in this region. Mousavinezhad et al. (2021) utilized a multiple linear regression model to disentangle the contributions of meteorology and emissions to ozone levels in North China during 2015-2019. Their findings indicated that meteorological factors such as increased downward surface solar radiation and near-surface air temperatures accounted for 32% of the observed ozone increase, while changes in emission precursors contributed 68%. To elucidate the interannual variability of ozone in North China, Gong et al. (2020) employed tagged $O_3$ simulations with the Goddard Earth Observing System Chemical Transport Model (GEOS-Chem) model and suggested that one-third of the rise in ozone pollution days observed from 2014 to 2018, particularly in 2018, could be attributed to emissions transport from central-eastern China. Considering the intertwined roles of meteorology and emissions, the focus shifted to examining ozone anomalies relative to their respective monthly averages, thereby minimizing the influence of emissions on ozone variability.

The study focuses on two specific regions—North China and central eastern China—to analyze days where regional mean MDA8 ozone levels exceeded 10 ppbv of their respective monthly means, defined as regional ozone pollution events. Observational data indicate a total of 131 and 89 such events in North China and central eastern China, respectively, during the summers of 2015-2019. Ozone pollution events are observed to extend meridionally (Figs. 5,6), northward into northeastern China from North China (Fig. 5a,b) and covering large areas of northern and southern China from



central eastern China (Fig. 6a,b).

During regional ozone pollution events, concurrent meteorological conditions
typically feature higher downward surface solar radiation, 2-meter air temperatures,
reduced water vapor, and decreased total cloud cover, all of which favored ozone
accumulation. Meteorological anomalies for each day are computed relative to their
respective months, with the study testing four different methods for deriving
climatology, including averages from the same day, same month, summer periods from
2015-2019, and summers from 1990-2019. They all yield comparable results.
Analyzing atmospheric blocking, we find that 43% (56 events) of regional ozone
pollution events in North China and 48% (43 events) in central eastern China are
accompanied by blocking. Notably, among the 36 events where ozone pollution
concurrently affected both North China and central eastern China, nearly 40% are
associated with blocking events.

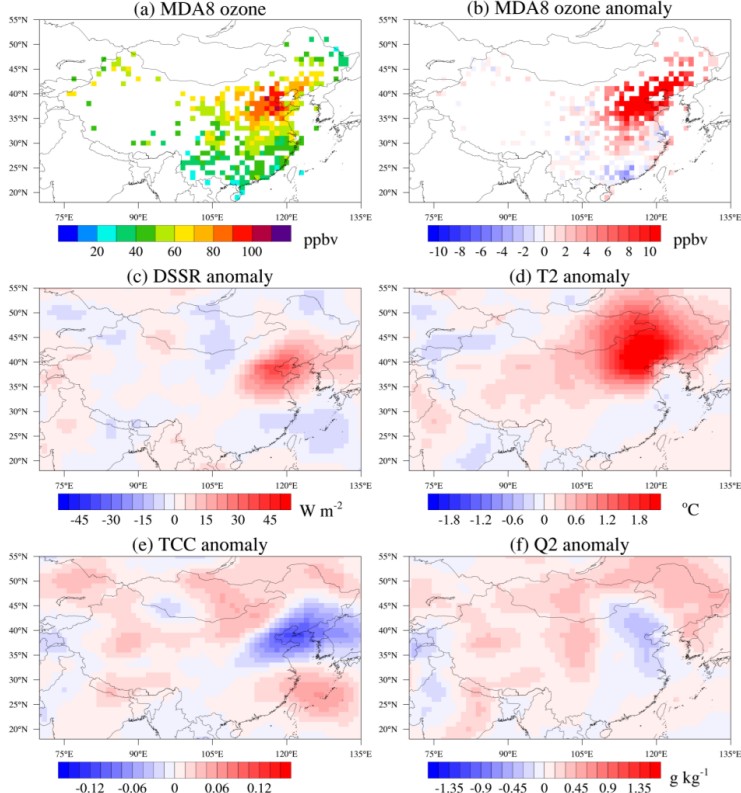

**Fig. 5 Spatial distributions of ozone and meteorological conditions during ozone pollution events in North China.** Shown are composited results of (a) mean MDA8 ozone concentrations, anomalies of (b) MDA8 ozone, (c) downward surface solar radiation, (d) 2-m air temperature, (e) total cloud cover, and (f) 2-m specific humidity during the summers of 2015-2019.





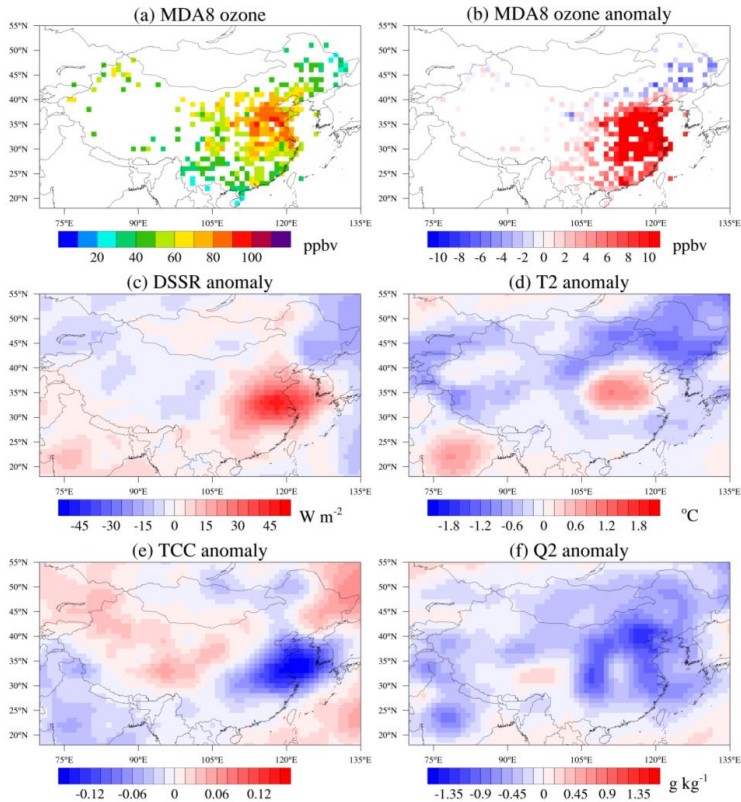

**Fig. 6 Spatial distributions of ozone and meteorological conditions during ozone pollution events in central eastern China.** Shown are composited results of (a) mean MDA8 ozone concentrations, anomalies of (b) MDA8 ozone, (c) downward surface solar radiation, (d) 2-m air temperature, (e) total cloud cover, and (f) 2-m specific humidity during the summers of 2015-2019.

The impact of blocking events on downstream meteorological conditions and ozone pollution is examined, primarily based on Rossby wave propagation, which profoundly affects large-scale circulations. For example, Ding and Li (2017) analyzed reanalysis data from 1951–2015 and found that Rossby waves originating from northwest Europe entered the North Africa-Asia westerly jet in the upper troposphere, propagating eastward along the subtropical westerly jet. This circulation favored persistent heavy rainfall events in South China (20°-30°N). Liu et al. (2022) studied data from 1979–



2020 and observed positive anomalies in summer shortwave cloud radiative effects over
northern Russia, promoting the generation of Ural blocking. This blocking dynamically
triggered a positive Eurasian pattern characterized by a "$+ - +$" wave train, resulting in
positive precipitation anomalies in northern China and strong heatwaves in southern
China. In addition to northwest Europe and northern Russia, blocking also occurs over
northeastern Russia. This, combined with the land-sea temperature contrast between
warm northeastern Eurasia and the colder Oyashio region in the North Pacific, may
induce a north–south-tilting anticyclone, leading to increased temperatures across a
wide area of China (Amano et al., 2023).
Blocking events are categorized into Euro-Atlantic, northern Russia, and North
Pacific regions (Fig. 7), based on their geographical locations. Analysis of NCEP
reanalysis data during the summers of 2015-2019 identified a total of 227 blocking days
in the Northern Hemisphere, with approximately 50% occurrence. Of these, 60 days
occurred over the Euro-Atlantic sector, 68 days over northern Russia, and 162 days over
the North Pacific. The higher frequency of blocking in the North Pacific is partly due
to conducive conditions in northeastern Russia and Alaska. Notably, the sum of
blocking events across these regions exceeds the total for the Northern Hemisphere,
owing to concurrent events in multiple areas. High blocking frequency has previously
been reported (Lupo, 2021), indicating climatologically in the Northern Hemisphere
there are 30-35 blocking events per year with a mean duration of 9 days. This
occurrence rate is higher than in our study, partly due to the larger frequency in winter
and fall compared to summer.
Anomalies of 500 hPa geopotential height from reanalysis data and MDA8 ozone
from observations during composite blocking events over Euro-Atlantic, northern
Russia, and North Pacific are depicted in Fig. 7. These illustrations highlight the
characteristics of Rossby wave propagation and the corresponding variations in ozone.
For instance, when blocking occurs over the Euro-Atlantic (top of Fig. 7), it coincides
with anomalously high pressure, triggering a wave number of 5 and resulting in high
pressure over northern China. This configuration leads to high ozone anomalies over



northeastern China, with scattered spots of high ozone anomalies over parts of North
China and central eastern China. When blocking shifts eastward to northern Russia
(middle row in Fig. 7), a positive Eurasian pattern emerges with a "+ − +" wave train.
This pattern manifests in negative anomalies in the northern flank of China and positive
pressure anomalies in central to southern eastern China, South Korea, and southern
Japan. During blocking over the North Pacific, spanning northeastern Russia and
Alaska (Fig. 7e), broad positive anomalies are observed in southern China. However,
notable anomalies of 500 hPa geopotential height are absent in southern China, and
positive high pressure is not always accompanied during ozone pollution events (Yang
et al., 2024).

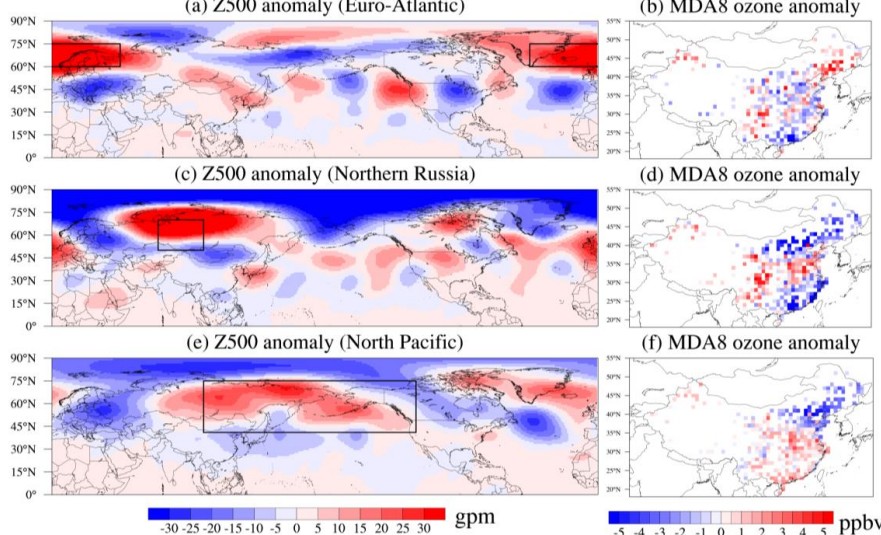


**Fig. 7 Spatial distributions of anomalies in 500 hPa geopotential height (gpm) and
ozone.** Shown are composited results during blocking events over Euro-Atlantic sector
(top), northern Russia (middle), and the North Pacific (bottom), indicated by the black
square.


To further elucidate the pathway of Rossby wave propagation, we focus on a typical
blocking event from June 27 to July 4, 2019. During this period, a blocking high is
situated over northern Russia and the eastern flank of the Ural Mountains (Fig. 8a).





Coincidentally, another blocking event (June 29 - July 4, 2019) occurs over the North
Pacific near Alaska. regions with convergence of wave activity flux indicate weakened
westerlies, suggesting an incoming wave train and accumulation of wave activity in
these areas. This accumulation could further amplify the blocking high (Nakamura et
al., 1997; Schneidereit et al., 2012), serving as a source region for Rossby wave
propagation.
A strong high-pressure system over northern Russia (Fig. 8a), propagating
southeastward (arrows in Fig. 8d). This propagation stimulates positive height
anomalies over central eastern China, evident in both the upper (200 hPa; Fig. 8b) and
mid-troposphere (500 hPa; Fig. 8d), with a weaker signal observed at the lower
troposphere (850 hPa; Fig. 8c), indicating a barotropic structure (Barriopedro et al.,
2006; Sui et al., 2022). The blocking events over northern Russia may originate from
the North Atlantic, as indicated by (Liu et al., 2022). This is suggested by the presence
of a positive geopotential height anomaly over the northern North Atlantic, which then
propagates northeastward towards northern Europe and Russia. This pattern resembles
the Rossby wave train with a zonal wavenumber of 5, as described in Xu et al. (2019).
It originates west of the British Isles and propagates towards Lake Baikal, simulating a
high-pressure system on the southern flank of China. The blocking over Alaska serves
as another source of Rossby waves, propagating eastward towards the Atlantic and
triggering another pathway through the Mediterranean Sea along the subtropical jet.
This process further enhances high-pressure anomalies over central eastern China (Fig.
8d).
Modulated by this large-scale circulation, there is an increase in downward surface
solar radiation, 2-m air temperature, reduced water vapor, and total cloud cover over
areas spanning 25° to 40°N (Fig. 8g-j). These conditions contribute to widespread
ozone increases in this region, extending slightly into North China and southern China
(Fig. 8e,f). Comparably, when atmospheric blocking occurs over Euro-Atlantic region,
a Rossby wave propagates southeastward from the northern Atlantic. This triggers high
pressure anomalies in North China and central eastern China, creating meteorological



conditions that favor anomalously high ozone concentrations (July 20 - 24, 2017, Fig.
S6). Additionally, a concurrent blocking event over the North Pacific initiates another
Rossby wave propagation, which converges with the Rossby wave originating from the
Euro-Atlantic blocking. This convergence reinforces the eastward propagation of the
Rossby wave.

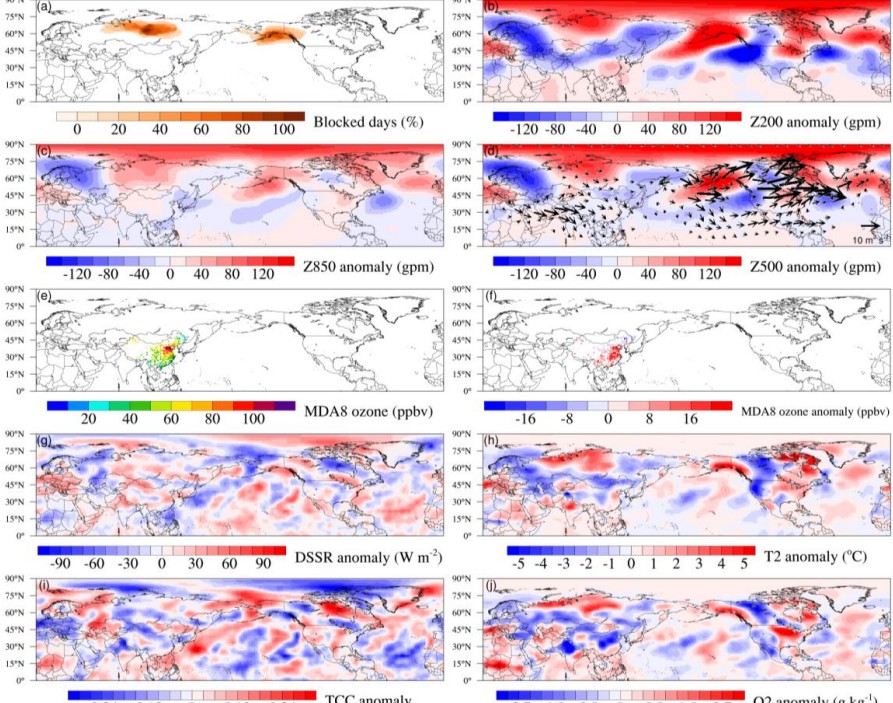


**Fig. 8 Spatial distribution of blocking, ozone and geopotential height.** Shown are
results of anomalies of geopotential height at (b), 200 hPa, (c) 850 hPa, (d) 500 hPa, (e)
ozone concentrations, anomalies of (f) ozone, (g), DSSR, (h) 2-m air temperature, (i)
total cloud cover and (j) 2-m specific humidity. The results are composited during a
specific blocking event over northern Russia.

Next, we explore how atmospheric blocking influences ozone through the effect of
BVOC emissions. In a previous study, significant improvements in summer blocking
simulations were achieved by increasing horizontal resolution in an Earth system model





with coupled atmosphere and ocean components (Gao et al., in review-a). Driven by
the prescribed SST, high-resolution simulations have shown enhanced blocking
frequencies, particularly over the Ural Mountains and North Pacific (Fig. S7).
Therefore, the analysis below is based solely on SW-HRESM.

We composite blocking events occurring over the Euro-Atlantic sector (100 days),

northern Russia (47 days) and North Pacific (119 days), and the spatial distribution of
ozone concentrations is shown in Fig. 9. The probability distribution function of ozone
concentrations is shown in Fig. 10. Several distinctive features emerge. During non-
blocking periods (Fig. S8a; Fig. 10), the mean ozone concentrations over North China
is slightly higher (66.3 ppbv) than in central eastern China (63.3 ppbv). Among all three
blocking categories, ozone concentrations over central eastern China tend to increase
to a larger extent compared to North China, resulting in comparable or higher ozone
concentration in central eastern China relative to North China (Fig. 9d,e,f). Specifically,
blocking triggers an ozone increases of 10.7 ppbv, 7.1 and 5.9 ppbv when blocking
occurs in the Euro-Atlantic, northern Russia and North Pacific sectors, respectively,
compared to values of 4.9 ppbv, 4.2 ppbv and 2.1 ppbv in North China (Fig. 10). When
blocking occurs in northern Russia and the North Pacific, the effect can extend further
south from central to southeastern China. Accompanied by the blocking, an increase in
downward surface solar radiation, 2-m air temperature, along with reduced water vapor,
and total cloud cover, emerges primarily over North China and central eastern China
(Fig. S9). Despite slight differences, this feature is consistent with the observed patterns
(Fig. 7b,d,f).

BVOC emissions play important roles in modulating ozone concentrations. When

the blocking occurs, the effects of BVOC emissions on ozone concentrations range
from 10.6 ppbv to 15.5 ppbv over North China and central eastern China (Fig. 9g,h,i;
Fig. 10), with the largest effect when blocking occurs over the Euro-Atlantic sector.
Consistent with the previous discussion on heatwaves (section 3.3), BVOC emissions
play a role even in the absence of blocking (Fig. S8b), with effects of 10.8 ppbv over
North China and 13.3 ppbv over central eastern China. The effect of BVOC emission



594 on ozone during blocking is larger than during non-blocking for most cases, except over

595 central eastern China during blocking in northern Russia, which is visible when

596 blocking is compared to a lower temperature range (i.e., < 26℃; Fig. S10). Overall, the

597 incremental effect of BVOC emissions on ozone during blocking, similar to that defined

598 in section 3.3, is calculated, and it could reach account for as much as 65% of the ozone

599 increase during blocking in North China 31% of the ozone increase during blocking in

600 central eastern China (Fig. 9j,k,l vs. Fig. 9g,h,i; Fig. 10).

601

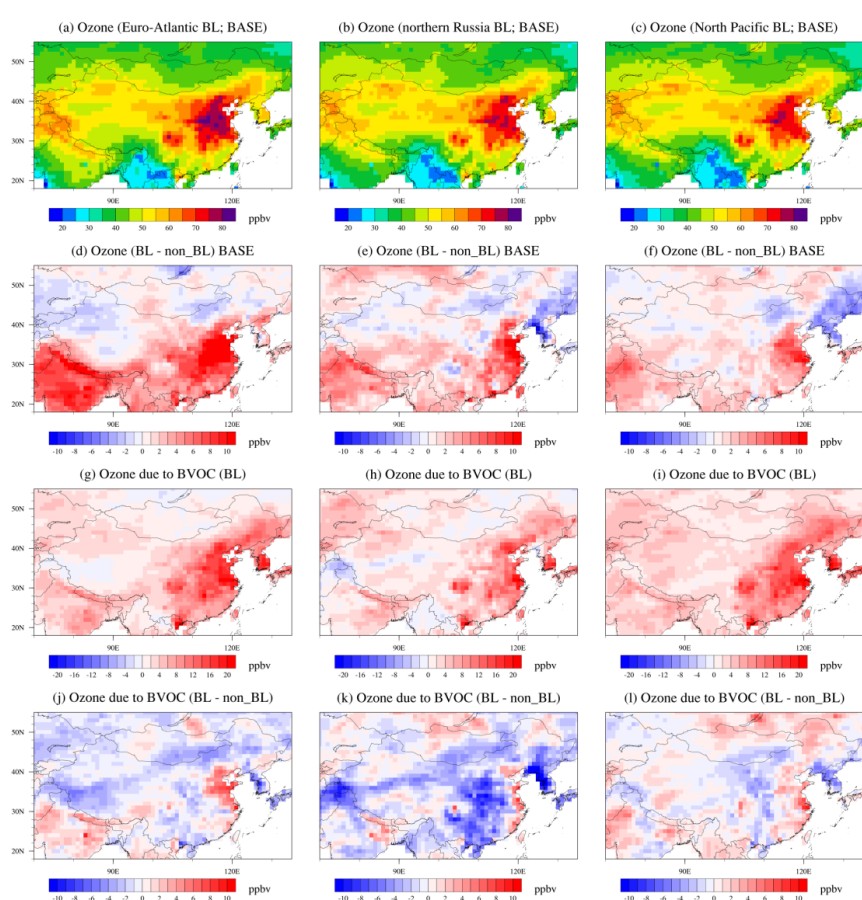

603 **Fig. 9 Spatial distributions of ozone concentrations.** Shown are results during

604 blocking over Euro-Atlantic (left column), northern Russia (middle column) and North

605 Pacific (right column) for (a,b,c) BASE, (d,e,f) ozone difference between blocking and



non-blocking, (g,h,i) effect of BVOC emissions, (j,k,l) differences of effects of BVOC
emissions on ozone between blocking and non-blocking.

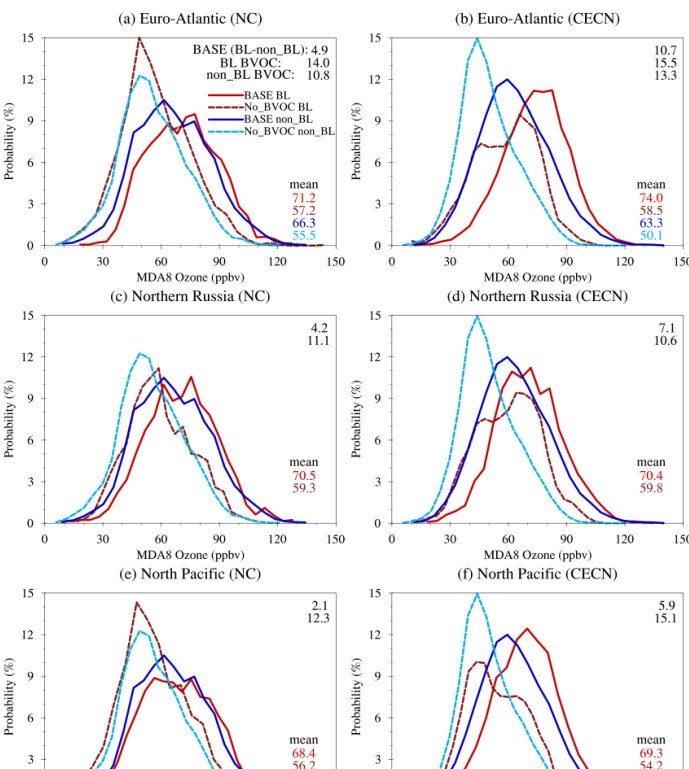


**Fig. 10 Probability distribution function of MDA8 ozone concentrations.** Shown
are results over North China (NC; left column) and central eastern China (CECN; right
column) during blocking events occurred at Euro-Atlantic sector (top), northern Russia
(middle) and North Pacific (bottom). The numbers on the top right of each panel
represent the MDA8 ozone enhancement between blocking and non-blocking (BASE
(BL-non_BL)), effect of BVOC emissions during blocking (BL BVOC) and non-
blocking (non_BL BVOC). The numbers on the bottom right of each panel show the
mean MDA8 ozone concentrations during blocking (in red) and non-blocking (in blue)
for BASE and the case without BVOC emissions. Since ozone values in the non-
blocking case is the same no matter where the blocking is, values for the non-blocking



case are only listed on the top row. The solid and dashed blue lines are the same between
middle, bottom rows and the top row.

**Conclusions**
Through the combination of high-resolution Earth system models and observations,
the effects of local meteorology and large-scale circulation on ozone concentrations are
elucidated. Based on observations and focusing on eastern China, we identify that
ozone pollution events are accompanied by anomalously high near-surface air
temperature, increased downward surface solar radiation, reduced water vapor and
decreased total cloud cover. We further find that blocking events over the Euro-Atlantic
sector, northern Russia and the North Pacific behave differently in modulating ozone
pollution in eastern China, controlled by the pathways of Rossby wave propagation.
While blocking in all three regions plays the most significant role in central eastern
China, blocking over northern Russia and the North Pacific may also impact the
southern part of China. Over the North Pacific, the large high-pressure system seems to
form a saddle-like structure, affecting widespread areas in southern China.
Moreover, blocking events could substantially trigger BVOC emission increases
and aggravate ozone pollution. Numerical experiments reveal that under favorable
meteorological conditions, such as heatwaves, BVOC emissions could play an even
larger role in triggering ozone increases, particularly in areas with lower anthropogenic
emissions. This highlights a potentially more critical role for BVOC emissions,
especially when anthropogenic emissions are projected to decrease. This is the first
attempt to link atmospheric blocking, BVOC emissions, and ozone pollution, which has
important implications for future studies, particularly those associated with the
mechanisms of how large-scale circulations affect ozone concentrations under a
warming climate.





**Data availability.** The CESM model output data are available from the iHESP data
portal (https://ihesp.github.io/archive/products/ihesp-products/data-
release/DataRelease_Phase2.html).
**Author contributions**
Y.G. conceived the project and designed the method, W.K. performed the analysis and
drafted the manuscript, X.G., X.A. helped on the analysis, D.T, W.L., M.C., X.G., S.Z.,
H.G., L.W.  helped on the interpretation of the results. All authors contributed to the
writing of the manuscript.

**Competing interests**
The authors declare that they have no conflicts of interest.

**Acknowledgements**
This work was supported by the National Natural Science Foundation of China
(42122039, 42375189), the Science and Technology Innovation Project of Laoshan
Laboratory (LSKJ202300401, LSKJ202202201) and Hainan Provincial Joint Project of
Sanya Yazhou Bay Science and Technology City (2021JJLH0050).

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
