# Peer review of "Enhanced understanding of atmospheric blocking modulation on ozone"

_EGUsphere, 2024_

## Author Comment (AC2)

**Response to Reviewer #1** (our response in **blue**)

We thank the reviewer for the comprehensive and detailed comments to help us further improve the manuscript. Please see the detailed responses to your comments below.

egusphere-2024-2500 presents a detailed model study into the role of meteorological controls on ozone pollution events and how meteorology can enhance ozone production through large-scale atmospheric circulation and heat waves.

I believe that this study will be suitable for publication in this journal with the following points addressed.

1. As the authors mention - $O_3$ production is dependent on both NOx emissions and VOC emissions (of which BVOC emissions are a large component). Many studies have considered $O_3$ production to be limited by either the availability of NOx or the availability of VOC's. Much of the world is NOx limited for $O_3$ production. $HO_2$ uptake onto aerosol has also been shown to inhibit $O_3$ production, with this being particularly important in Eastern China (https://www.nature.com/articles/s41561-022-00972-9). I think further discussion of the "limits" on $O_3$ production would be beneficial to this study, particularly if the increased BVOC emissions caused by heatwaves etc occur in a NOx-limited environment.

Thank you for the comment. We agree with the reviewer that the majority of the world is NOx-limited for ozone production. Meanwhile, in highly polluted urban and industrial areas, VOC-limited conditions are also quite common. In this study, we primarily focus on highly polluted urban regions, where increased BVOC emissions typically enhance ozone formation. Regarding NOx-limited regions, the positive effect of BVOC on ozone can also be observed, with one of the key reasons being the influence of transport. In one of our previous studies (Wang et al., 2022), taking Shandong Province as an example, we found that although BVOC emissions in Shandong were relatively low, the impact of BVOC on ozone levels in the province could exceed 10 ppbv. Sensitivity numerical experiments revealed that biogenic emissions from southern China led to an increase in ozone, which was subsequently

transported northeastward, contributing to the elevated ozone concentrations in Shandong.

Regarding the aerosol uptake of $HO_2$ and its suppressive effect on ozone formation, we carefully examined the literature provided by the reviewer. In urban areas with relatively sufficient NOx but insufficient VOCs, the reaction between VOCs and OH generally has a greater impact on ozone than the effect of aerosol uptake. In regions with relatively insufficient NOx and high aerosols, such as parts of eastern China, the aerosol uptake of $HO_2$ may also become the dominant mechanism driving changes in ozone concentration, as illustrated in Figure 1 of the literature mentioned by the reviewer. In this case, when VOCs are added to the numerical model, the ozone changes likely reflect the net effect. Future efforts to isolate this effect would help further understand the mechanisms and sources of BVOC impacts on ozone. We have incorporated this information, including the effects of transport and aerosol uptake on ozone, into the last section (Discussion) of the revised manuscript.

Reference:

Wang, H., Gao, Y., Sheng, L., Wang, Y., Zeng, X., Kou, W., Ma, M., and Cheng, W.: The Impact of Meteorology and Emissions on Surface Ozone in Shandong Province, China, during Summer 2014–2019, Int. J. Environ. Res. Public. Health, 19, 6758, 10.3390/ijerph19116758, 2022.

2. Whilst this study focuses on BVOC emissions - Halogens play a large role in tropospheric Ox loss, particularly iodine with the global loss of O3 due to iodine being comparable to the production of O3 through isoprene (eg Alicke et al 1999, Saiz-Lopez et al 2012, Sherwen et al 2016, Pound et al 2023). Does the chemistry scheme in this study include halogens? If not do the authors believe that periods of high O3 could be impacted by including halogen chemistry?

We agree that halogens play an important role in affecting tropospheric ozone concentrations. Based on the reviewer's suggestion, we have conducted preliminary tests using Community Earth System Model (CESM) version 2.2, referencing relevant

studies (e.g., Badia et al., 2021; Li et al., 2022; Saiz-Lopez et al., 2023). The atmospheric and land components are CAM6 and CLM5, respectively. The ocean and sea ice data are prescribed from the Merged Hadley-NOAA/OI Sea Surface Temperature & Sea-Ice Concentration dataset (Hurrell et al., 2008). To improve the accuracy of meteorological simulations, we applied the nudging method. The reanalysis data used is the 6-hourly reanalysis dataset from the Modern-Era Retrospective Analysis for Research and Applications, Version 2 (MERRA-2). The variables nudged include air temperature, eastward wind, and northward wind. We selected the period from January to December 2019 as the study period (with a spin-up time of six months). The model was configured with low-resolution (~1°) simulations. The emissions are based on the Community Emissions Data System (CEDS) emission dataset under the SSP370 scenario, and two sets of numerical simulations were conducted: one without complex halogen chemistry and the other with complex halogen chemistry included (details on halogen reactions in Saiz-Lopez et al., 2023).

Figure 1 shows that compared to the case without halogen chemistry, the inclusion of halogen chemistry substantially reduces the simulated ozone concentrations for the U.S., Europe, and Eastern China. Relative to the observations, when examining this single year of simulation, the average bias is reduced from 10% (for the three regions, 13% 10% 7%) to 3% (6% 2% 2%). Most of the improvement occurs in winter, spring, and fall, while changes during summer are relatively smaller.

However, note that this is only a one-year simulation, and more future work is needed to fully examine the effectiveness of halogen chemistry on ozone. For instance, useful tests include firstly the simulations of multi-year instead of only one-year. Secondly, it is useful to conduct high-resolution Earth system simulations (e.g., 25 km) to take advantage of finer resolution emissions and the spatial heterogeneities in emissions. Thirdly, this study primarily focuses on large regional scales; future evaluations can further assess simulations over smaller regions and specific ozone pollution episodes.

The above tests were conducted for Cl, Br, and I. In the future, individual halogens, such as iodine, could be tested separately. Iodine serves as an important ozone sink (Alicke et al., 1999; Pound et al., 2023; Saiz-Lopez et al., 2012; Sherwen et al., 2016).

Based on Sherwen et al. (2016), the impact of iodine on ozone is primarily observed over tropical oceans, with relatively limited effects on near-surface ozone. More information can be investigated in future studies.

The points discussed above have been further elaborated at the end of the Discussion section.

[Figure]

Fig. 1 Evaluation of MDA8 ozone in 2019 for the U.S., Europe, and Eastern China using CESM simulations: comparison of cases with and without halogen chemistry

References:

Alicke, B., Hebestreit, K., Stutz, J., and Platt, U.: Iodine oxide in the marine boundary layer, Nature, 397, 572-573, 10.1038/17508, 1999.

Badia, A., Iglesias-Suarez, F., Fernandez, R. P., Cuevas, C. A., Kinnison, D. E., Lamarque, J.-F., Griffiths, P. T., Tarasick, D. W., Liu, J., and Saiz-Lopez, A.: The Role of Natural Halogens in Global Tropospheric Ozone Chemistry and Budget Under Different 21st Century Climate Scenarios, Journal of Geophysical Research: Atmospheres, 126, e2021JD034859, 10.1029/2021JD034859, 2021.

Hurrell, J. W., Hack, J. J., Shea, D., Caron, J. M., and Rosinski, J.: A New Sea Surface Temperature and Sea Ice Boundary Dataset for the Community Atmosphere Model, J Climate, 21, 5145-5153, 10.1175/2008JCLI2292.1, 2008.

Li, Q., Fernandez, R. P., Hossaini, R., Iglesias-Suarez, F., Cuevas, C. A., Apel, E. C., Kinnison, D. E., Lamarque, J.-F., and Saiz-Lopez, A.: Reactive halogens increase the global methane lifetime and radiative forcing in the 21st century, Nat. Commun., 13, 2768, 10.1038/s41467-022-30456-8, 2022.

Pound, R. J., Durcan, D. P., Evans, M. J., and Carpenter, L. J.: Comparing the Importance of Iodine and Isoprene on Tropospheric Photochemistry, Geophys. Res. Lett., 50, e2022GL100997, 10.1029/2022GL100997, 2023.

Saiz-Lopez, A., Plane, J. M. C., Baker, A. R., Carpenter, L. J., von Glasow, R., Gómez Martín, J. C., McFiggans, G., and Saunders, R. W.: Atmospheric Chemistry of Iodine, Chem. Rev., 112, 1773-1804, 10.1021/cr200029u, 2012.

Saiz-Lopez, A., Fernandez, R. P., Li, Q., Cuevas, C. A., Fu, X., Kinnison, D. E., Tilmes, S., Mahajan, A. S., Gómez Martín, J. C., Iglesias-Suarez, F., Hossaini, R., Plane, J. M. C., Myhre, G., and Lamarque, J.-F.: Natural short-lived halogens exert an indirect

cooling effect on climate, Nature, 618, 967-973, 10.1038/s41586-023-06119-z, 2023.

Sherwen, T., Schmidt, J. A., Evans, M. J., Carpenter, L. J., Großmann, K., Eastham, S. D., Jacob, D. J., Dix, B., Koenig, T. K., Sinreich, R., Ortega, I., Volkamer, R., Saiz-Lopez, A., Prados-Roman, C., Mahajan, A. S., and Ordóñez, C.: Global impacts of tropospheric halogens (Cl, Br, I) on oxidants and composition in GEOS-Chem, Atmos. Chem. Phys., 16, 12239-12271, 10.5194/acp-16-12239-2016, 2016.

3. (Lines 160-162) Tropospheric ozone has a lifetime of approximately 23 days as a global average (Young et al 2013), do the authors believe that a one-month spinup is sufficient to remove the influence of initial conditions, many modelling studies into ozone tend to use periods of 1year or more.

Thank you for the comment. Many studies use more than a year of spin-up time, primarily because stratospheric ozone equilibrates more slowly and typically requires a longer period to reach a stable state. However, our study does not involve stratospheric chemistry, as stratospheric ozone is prescribed. Therefore, for tropospheric ozone, a spin-up of one to a few months should be sufficient to achieve equilibrium.

In our simulation, we indeed applied an extra six-month spin-up period for ozone, followed by a one-month spin-up for each numerical experiment. In the revised manuscript, we have modified this section of the description.

4. Section 3.1 and others contain a large number of statistics about the ozone events being studied (particularly lines 245-258). I believe this information could be better suited to being summarised in a table for clarity.

Based on the reviewer's suggestion, we have incorporated this information into Table 1, as shown below (lines 245-258 in the manuscript), and made corresponding revisions to the related description.

Table 1. Regional mean ozone exceedance rates (%) during 2015-2019

| Region | Annual 4th | | | All days | | Peak season | |
|---|---|---|---|---|---|---|---|
| | 51 ppbv | 61 ppbv | 82 ppbv | 51 ppbv | 61 ppbv | 36 ppbv | 51 ppbv |
| U.S. | 98 | 78 | 2 | 17 | 4 | 98 | 15 |
| Europe | 89 | 60 | 4 | 11 | 2 | 89 | 8 |
| China | 99 | 96 | 77 | 31 | 18 | 96 | 60 |

5. Section 3.2 (lines 277-278) - The authors discuss the differences in BVOC emission between the two model resolutions. What do the authors believe is the cause of this difference and is there any possible method to attempt to control for this and remove model resolution as a potential factor in emission sensitivity. Other models have concluded that emission calculations are sensitive to the resolution of meteorology used to calculate them and as such use a pre-calculated emission at a fixed spatial resolution see doi:10.1038/s41597-020-0488-5

We thank the review for raising the resolution dependence of meteorology, which may further affect the BVOC emissions. The reference provided by the reviewer has been cited in the submitted manuscript, and we have noticed this study, and many of other chemical transport models may offline simulate BVOCs. However, the focus of this study is somewhat different. As meteorological conditions significantly influence BVOCs, one of the objectives in improving and applying high-resolution Earth system models is to investigate how these models enhance the simulation capability and mechanisms of meteorological conditions, particularly extreme weather events. Based on the improved meteorological conditions, the aim is to achieve more accurate BVOC emissions and ozone simulation precision. For example, in Section 3.2, we discuss the differences in BVOC emissions between two model resolutions. We believe this difference primarily arises from the varying accuracy in the simulation of meteorological variables (such as temperature, radiation, and humidity), as well as the level of detail in the representation of topography at different resolutions. At the same time, using high-resolution models allows for a better distinction of the spatial variability in BVOC emissions, thereby facilitating a more accurate assessment of their impact on atmospheric pollutants. We also agree with the reviewer's perspective that the differences in BVOC emissions between the high-resolution and low-resolution models discussed in this study arise from both resolution differences and meteorological differences. This study did not separate these factors, and we have included this limitation in the final section (Discussion) of the manuscript.